# Real-World Outcomes for Localised Gastro-Oesophageal Adenocarcinoma Cancer Treated with Perioperative FLOT and Prophylactic GCSF Support in a Single Asian Centre

**DOI:** 10.3390/cancers16213697

**Published:** 2024-11-01

**Authors:** Wanyi Kee, Kennedy Yao Yi Ng, Shun Zi Liong, Siqin Zhou, Sharon Keman Chee, Chiew Woon Lim, Justina Yick Ching Lam, Jeremy Tian Hui Tan, Hock Soo Ong, Weng Hoong Chan, Eugene Kee Wee Lim, Chin Hong Lim, Alvin Kim Hock Eng, Christabel Jing Zhi Lee, Matthew Chau Hsien Ng

**Affiliations:** 1Division of Medical Oncology, National Cancer Centre Singapore, 30 Hospital Blvd, Singapore 168583, Singaporechristabel.lee.j.z@nccs.com.sg (C.J.Z.L.); 2Oncology Academic Clinical Programme, Duke-NUS Medical School, Singapore 169857, Singapore; 3Division of Clinical Trials and Epidemiology, National Cancer Centre Singapore, Singapore 168583, Singapore; 4Division of Surgery and Surgical Oncology, Singapore General Hospital, Singapore 169608, Singaporechan.weng.hoong@singhealth.com.sg (W.H.C.); eugene.lim.k.w@singhealth.com.sg (E.K.W.L.);; 5Division of Surgery and Surgical Oncology, National Cancer Centre Singapore, Singapore 168583, Singapore

**Keywords:** FLOT, gastric, gastroesophageal, perioperative chemotherapy

## Abstract

**Simple Summary:**

Perioperative FLOT (5-fluorouracil, oxaliplatin and docetaxel) is a standard of care for patients with locally advanced gastro-oesophageal adenocarcinoma (GEA) in Western guidelines, but its use is limited in Asian patients. The aim of our retrospective study was to assess the safety and efficacy of perioperative FLOT in Asian patients from a single centre with routine granulocyte colony-stimulating factor (GCSF) prophylaxis. We demonstrate similar tolerability and efficacy for FLOT to that reported in Western populations but with lower rates of grade 3 to 4 neutropenia. Elderly patients ≥ 65 years old had similar outcomes to those under 65. However, patients of lower socioeconomic status were more likely to experience severe AEs, highlighting the need to proactively support vulnerable groups during treatment.

**Abstract:**

Background: Perioperative FLOT (5-fluorouracil, oxaliplatin and docetaxel) is a standard of care for patients with locally advanced gastro-oesophageal adenocarcinoma (GEA) in Western guidelines, but its use is limited in Asian patients. We report outcomes from a single Asian centre of perioperative FLOT with concomitant granulocyte colony-stimulating factor (GCSF) prophylaxis. Methods: A retrospective analysis of all 56 stage II to III GEA patients treated with perioperative FLOT at the National Cancer Centre Singapore between June 2017 and February 2024 was performed. All patients were discussed at a multidisciplinary tumour board, underwent preoperative laparoscopic staging, and received prophylactic GCSF with perioperative FLOT. Surgery was performed across four partner institutions. The primary endpoints were the tolerability of FLOT and pathological complete response (pCR). A univariate analysis of factors associated with survival and adverse events was also performed. Results: Overall, 33 patients (58.9%) completed eight cycles of pre- and postoperative FLOT, and 92.9% underwent resection. The commonest grade 3 to 4 adverse events (AEs) were diarrhoea (10.7%) and neutropenia (5.6%). The 30- and 90-day postoperative mortality rates were 0% and 1.9%, respectively. In resected tumours, the pCR was 15.4%. The median DFS was 27.5 months, but the median OS was not reached. The values for 1-, 2-, and 3-year DFS were 74.6%, 61.0%, and 46.5%, respectively. The values for 1-, 2-, and 3-year OS were 85.0%, 67.4%, and 61.0%, respectively. In the univariate analysis of patients who underwent resection, an ECOG status of 0 was associated with better DFS, while ypN0, R0 resection, and pathological stages 0-II were associated with better DFS and OS. Patients ≥ 65 years benefited from FLOT similarly to those <65 years in terms of DFS (HR 1.03; *p* = 0.940) and OS (HR 1.08; *p* = 0.869), with similar rates of grade 3 to 4 AEs. Patients with a higher housing index (HI) were less likely to experience ≥grade 3 AEs compared to those with a lower HI (OR 0.16, *p* = 0.029). Conclusions: This study presents a unique real-world Asian experience of perioperative FLOT with prophylactic GCSF use, with low rates of G3 to 4 neutropenia. The tolerability of FLOT was similar to that reported in Western populations. Furthermore, similar survival and rates of grade 3 to 4 AEs were observed in elderly patients. Patients of lower socioeconomic status were more likely to experience severe AEs, highlighting the need to proactively support vulnerable groups during treatment.

## 1. Background

Gastric cancer is the fifth most common cancer and the fourth leading cause of cancer-associated mortality worldwide [1]. In patients with resectable gastric (GC) or gastroesophageal junction (GEJ) adenocarcinoma, adjuvant chemoradiation [2] and adjuvant [3,4,5] and perioperative chemotherapy improved survival compared to surgery alone [6,7]. In the pivotal MAGIC trial, perioperative ECF chemotherapy significantly improved overall survival (OS) (hazard ratio [HR]: 0.75) and 5-year OS by 13% [6]. Similar results were seen in the FNLCC/FFCD trial using 5-fluorouracil plus cisplatin [7]. Subsequently, the FLOT4 study demonstrated superior pathological response, R0 resection rates, and survival with perioperative FLOT compared to ECF or ECX chemotherapy and has become the standard of care for locally advanced resectable gastric or gastro-oesophageal adenocarcinoma, particularly in Western guidelines [8,9]. More recently, the ESOPEC study has demonstrated the superiority of perioperative FLOT over preoperative chemoradiation with carboplatin and paclitaxel in resectable GEJ and esophageal adenocarcinomas [10].

Worldwide, there are geographical differences in treatment strategies for locally advanced resectable GC and GEJ cancer. In East Asian countries, upfront resection followed by adjuvant chemotherapy is the preferred approach, with preoperative chemotherapy reserved for tumours that are clinically T4 or with bulky nodes [11,12]. However, non-FLOT regimens such as DOS [13,14], SOX [15,16], and S1-cisplatin [11] are more commonly used. FLOT is associated with higher rates of grade 3 to 4 neutropenia and diarrhoea compared to ECF/ECX [17,18]. Studies of perioperative FLOT in Asian gastric cancer patients are limited [19,20,21]. Furthermore, docetaxel has also been reported to have higher rates of adverse events (AEs), such as neutropenia in Asian patients [22]. There is also concern over the tolerability of FLOT in elderly patients; in the FLOT4 study, only 24% of patients were ≥70 years old [18]. In comparison, when compared with doublet chemotherapy in the metastatic setting, elderly patients ≥65 years receiving modified FLOT in the GASTFOX study [23] did not appear to have an OS benefit, while in the FLOT65+ study [24], elderly patients ≥70 years receiving FLOT had no PFS benefit. Here, we present our experience of FLOT with prophylactic GCSF (granulocyte colony-stimulating factor) support in a single Asian centre and examine factors that may be associated with toxicity and efficacy.

## 2. Methods

### 2.1. Study Population

A retrospective review of 56 consecutive patients with resectable or borderline resectable gastro-oesophageal adenocarcinoma (GEA) treated with perioperative FLOT at the National Cancer Centre Singapore (NCCS) between June 2017 and February 2024 was performed.

### 2.2. Diagnosis and Treatment

Patients with histologically confirmed gastric and GEJ adenocarcinoma underwent preoperative staging with computed tomography (CT) of the thorax, abdomen, and pelvis or PET (positron emission tomography)/CT as well as preoperative laparoscopic staging with peritoneal washings. Clinicopathological staging was carried out in accordance with the 8th edition of the American Joint Committee on Cancer (AJCC) Tumour–Node–Metastasis (TNM) classification [25].

Primary tumour location was classified according to the Siewert classification for GEJ cancer. All patients received a perioperative FLOT regimen, as previously described [18], and prophylactic GCSF (Day 3–5 or pegylated Day 3). Dose modifications, delays, and treatment cessation were performed at the discretion of the primary physician.

All cases were discussed at multidisciplinary meetings, and surgery was undertaken across 4 hospitals, which were local to the patient. For gastric cancers, gastrectomy with D2 dissection was performed. For GEJ tumours, esophagectomy with 2-field nodal dissection was performed.

### 2.3. Data Collection and Study Endpoints

Clinical data collected from electronic medical records included demographic information, histology, stage at diagnosis, type of surgery, chemotherapy intensity, resection margin status, pathological staging and histology, mismatch repair (MMR) status, AEs, and tumour response grading (TRG). AEs were graded according to the National Cancer Institute Common Terminology Criteria for AEs version 5.0 [26]. TRG was determined using the National Comprehensive Cancer Network and College of American Pathologists (NCCN-CAP) grading system [8,27]. Socioeconomic status, using the housing index for each patient, was also analysed, as previously described [28].

Weight collected at 4 time points was analysed: at diagnosis, preoperatively, postoperatively, and post-treatment. Preoperative weight was the closest weight recorded before cycle 4 of preoperative FLOT and surgery. Postoperative weight was the closest weight recorded to the date of surgery. Post-treatment weight was the latest weight recorded prior to and up to 3 weeks after the last cycle of postoperative FLOT.

The primary endpoints of this study were the evaluation of pathological complete response (pCR; NCCN-CAP TRG 0) and the safety of perioperative FLOT.

The secondary endpoints were DFS and OS. Safety data, treatment compliance, and the correlation of clinical, demographic, and pathological factors with survival outcomes were analysed.

### 2.4. Statistical Analysis

OS was defined as the time from the date of initiation of preoperative FLOT treatment until the date of death from any cause or last follow-up. DFS was defined as the period from the date of initiation of preoperative FLOT treatment until the date of progression, relapse, death, or last follow-up. The median weight of patients at different phases of treatment were compared using the Wilcoxon signed rank test. Univariable logistic regression analyses were carried out to assess the relationship between the variables and the prevalence of AEs. OS and DFS were estimated with their corresponding two-sided 95% confidence interval (CI) using the Kaplan–Meier method. The reverse Kaplan–Meier method was used to estimate the median follow-up time and corresponding two-sided 95% CI. Univariable cox regression analysis was performed to assess the association between the variables and survival time. All *p*-values were two-sided, and those less than 0.05 were considered statistically significant. Statistical analyses were performed by using R software (version 4.2.0). The manuscript was written in accordance with the STROBE guidelines [29].

### 2.5. Ethics Approval

The SingHealth Central Institutional Review Board granted ethical approval for this study (2018/3046, 2023/2173, 2015/2363). Patient details were anonymised prior to data analysis. Written informed consent was obtained for patients who were alive. Consent was waived for patients who died or were lost to follow-up.

## 3. Results

### 3.1. Patient Characteristics

Between June 2017 and February 2024, a total of 56 patients received perioperative FLOT. Forty-seven patients (84%) were male, and 9 patients (16%) were female. The median age was 64 years (range: 37 to 74), and 44.6% and 14.3% of patients were ≥65 years and ≥70 years, respectively. There were more GEJ than gastric tumours (59% vs. 41%, respectively). All patients had an ECOG status of 0 or 1. The presence of signet ring cells was reported in 12 (21.4%) patients. At diagnosis, 91.1% (n = 51) patients had clinical stage T3-4, and 76.8% (n = 43) had nodal involvement. MMR status was reported in 76.8% (n = 43) of patients; 3.6% (n = 2) were dMMR, and 73.2% (n = 41) were pMMR. The baseline characteristics are shown in Table 1.

### 3.2. Treatment Exposure

Treatment exposure data are shown in Figure 1. Forty-eight patients (85.7%) completed four preoperative cycles of FLOT, of whom 39 patients (69.6%) did not require dose reductions. Thirty-seven patients (66.1%) received postoperative FLOT. Two patients received all eight cycles of FLOT preoperatively due to the resection margin being at risk or inadequate fitness for surgery. Thirty-three patients (58.9%) completed eight cycles of pre- and postoperative FLOT. Three patients received radiotherapy; two for R1 resection and one preoperatively for tumour bleeding. Appendix A summarises the data on perioperative FLOT. All patients received prophylactic GCSF; 35 (64.8%) patients received daily GCSF, 19 (33.6%) patients received pegylated GCSF, and 2 (3.6%) patients received both.

### 3.3. Safety

AEs are reported in Table 2. diarrhoea was the most commonly reported G3 to 4 AE at 10.7%, followed by neutropenia (5.4%) and vomiting (5.4%). Febrile neutropenia was reported in 3.6% of all patients. The commonest AEs overall were diarrhoea (46.4%), nausea (33.9%), and neuropathy (37.5%). Notably, there was no G3 to 4 neuropathy.

The values of 30-day and 90-day postoperative mortality were 0% and 1.9%, respectively. Postoperative complications of Clavien–Dindo grade 3 and above were reported in 25.0% of patients.

The univariate analysis of factors associated with AEs is shown in Appendix A, respectively. Patients with an ECOG status of 1 (odds ratio (OR) 0.14, *p* = 0.048) and node-negative disease (OR 0.05, *p* = 0.013) were less likely to experience AEs of all grades. Patients with a medium housing index were less likely to experience G3 and above AEs compared to those with a low housing index (OR 0.16, *p* = 0.029). There was no difference in G3 to 4 AEs between patients ≥65 years and <65 years (OR 0.28, *p* = 0.087).

### 3.4. Surgical and Pathological Findings

Of the 56 patients (92.9%), 52 underwent surgical resection. The reasons for not proceeding to surgery were the development of distant metastasis (n = 1), refusal of surgery (n = 1), sudden death (n = 1), and transfer of care to another institution (n = 1).

The surgery and pathology results are shown in Appendix A. Of the 52 patients who underwent surgery, the type of resection was total gastrectomy in 19 (36.5%), subtotal gastrectomy in 10 (19.2%), and esophagectomy in 20 (38.4%). The R0 resection rate was 75.0%. Pathological stages ypT0 and ypN0 were achieved in 15.4% and 42.3%, respectively, and pCR was achieved in 15.4%.

### 3.5. Weight Trend and Data

There was no significant weight loss during preoperative FLOT (Figure 2), but a significant decrease in median weight of 5.5 kg was observed between the preoperative and post-treatment phase (*p* < 0.001; Appendix A). Interestingly, there was no significant difference in the completion rate of all four cycles of postoperative FLOT in those with ≥ grade 2 compared with < grade 2 weight loss from baseline to the end of FLOT treatment (68.8% vs. 42.5%; *p* = 0.519).

### 3.6. Survival Outcomes

At the time of analysis, the median follow-up duration was 38.8 months. Fifteen patients died, and one patient was lost to follow-up. Of the patients who underwent surgical resection, 1 (1.9%) experienced local recurrence, 13 (25.0%) developed distant recurrence, and 3 (5.8%) had both local and distant recurrence.

The median DFS was 27.5 months (95% CI 22.5–47.6) (Figure 3). The 1-year, 2-year, and 3-year DFS rates were 74.6%, 61.0%, and 46.5%, respectively. Median OS was not reached (Figure 4). The 1-year, 2-year, and 3-year OS rates were 85.0%, 67.4%, and 61.0%, respectively.

In the univariate analysis, in patients who underwent resection, patients who achieved ypN0 had better DFS, with a HR of 0.10 (*p* < 0.01), and OS, with a HR of 0.14 (*p* = 0.011), compared to those with pathological node positivity. Patients with pathological stage III–IV disease had worse DFS (HR 3.04, *p* = 0.026) and OS (HR 3.63, *p* = 0.014) compared to patients with pathological stage 0-II disease (Appendix A). DFS and OS were better in patients who achieved R0 resection (DFS HR 0.34 [*p* = 0.045], OS HR 0.17 [*p* < 0.01]) compared to those with positive resection margins (Appendix A). DFS was worse in patients with an ECOG status of 1 compared to an ECOG status of 0 (HR 2.72 [*p* = 0.039]). OS was significantly longer in patients who had <G3 Clavien–Dindo complications (HR 0.30 [*p* = 0.022]) compared to those who with ≥G3. In contrast, there was no significant difference in survival between patients who achieved pCR and those who did not (DFS HR 0.28 [*p* = 0.220], OS HR 0.35 [*p* = 0.316]). There was also no difference in DFS (HR 1.03, *p* = 0.940) and OS (HR 1.08, *p* = 0.869) for patients <65 years and ≥65 years. The univariate analyses for DFS and OS are summarised in Appendix A and Table 3, respectively.

## 4. Discussion

Perioperative FLOT has become a standard of care for fit patients with locally advanced GEA in Western populations since the publication of the FLOT4-AIO study [8,9,17]. Since then, multiple studies on the real-world outcomes of perioperative FLOT in the Western population have been reported [30,31,32,33]. Some Asian countries have also evaluated the feasibility and tolerability of FLOT in the real-world setting, some of which utilised a modified FLOT regimen [19,20,21]. We present a unique Asian experience with prophylactic GCSF use to address the issue of higher rates of neutropenia with FLOT and docetaxel use in Asian patients [17,18,22].

In our centre, patients with ≥T3Nx GEJ and cT4 and/or bulky nodal burden gastric tumours are selected for perioperative chemotherapy, while less locally advanced tumours are resected upfront. This is reflected in the high rates of cT3/4 (91.1%) and cN+ (76.8%) tumours and R0 rate of 75%. Similarly, in the RESOLVE study, which selected for cT4 tumours [15], only 67% of patients in the perioperative chemotherapy group proceeded with the planned surgery, reflecting this challenging population. Nonetheless, our median DFS and 3-year OS rate are similar to those reported in the FLOT4 study [18], and our median OS is comparable to other Asian studies [20].

In our study, pCR was seen in 15.4% of resected tumours. This result is similar to that in the FLOT4-AIO study (16%) and NeoFLOT study (20.0%) [17,34] and higher than that in the MATTERHORN study (7%) [35]. Other real-world studies also report pCR rates of between 7.3% and 21.3% [30,32,36].

Treatment exposure in our study was comparable to the FLOT4-AIO study; 30.4% of our patients required dose reduction de-intensification, or discontinuation of FLOT in the preoperative phase compared to 26% in the FLOT4-AIO study [17]. The rate of completion of four and eight cycles of FLOT was also comparable to the FLOT4-AIO study [18].

Contrary to the FLOT4-AIO study, where the rate of G3 to 4 neutropenia was 52% [17], G3 to 4 neutropenia was reported in only 5.6% of our study population. This is also lower than other real-world studies, which report G3 to 4 neutropenia rates of 9.6% to 59.1% [19,20,30,32,36]. The rate of febrile neutropenia (3.4%) was also slightly lower than in the FLOT4-AIO study (5%) and other real-world studies (4.4–5.5%) [17,30,32]. This is likely due to the use of prophylactic GCSF for all patients. The most common G3 to 4 AE in our study was diarrhoea (10.7%), which was higher than in the FLOT4-AIO study (7%) [36]. Other real-world studies report G3-4 diarrhoea rates of 3.5% to 15.9% [19,20,30,32,36]. We also examined predictors of toxicity in our Asian population. Notably, we found that patients of higher socioeconomic status, as measured by housing index, were less likely to experience G3 and above AEs compared to those of lower status (OR 0.16, *p* = 0.029). This highlights the potential disparity in outcomes and tolerability of treatment in different socioeconomic groups as well as the importance of measures to support these vulnerable groups. A validated Singapore Housing Index (SHI) has been shown to have significant associations with survival outcomes in other forms of cancer as well in our local population [28,37]. While previous Western data suggest that socioeconomic deprivation is not associated with survival outcomes in the setting of curative oesophageal carcinoma surgery [38], other Western studies have shown that ethnic minorities have poorer outcomes [39,40]. These findings may not be applicable to all Asian countries due to different social support and healthcare funding systems.

In our study, the DFS, OS, and rates of G3-4 AEs were not significantly different between patients ≥65 years and <65 years. A similar OS benefit with FLOT was also seen in the AIO-FLOT4 study in patients ≥70 years and <70 years, suggesting that FLOT treatment can be considered in fit elderly Asian patients. In Asia, the DOS regimen, which has a lower docetaxel dose intensity, has also shown survival benefits when given in a preoperative setting in the PRODIGY study [13], and the ongoing JCOG2204 trial [41] is currently comparing these two neoadjuvant regimens. Similar regimens such as DOC have also been evaluated as preoperative regimens [42]

Significant changes in the median weight of our study population were noted in the postoperative phase of treatment. Studies have shown that significant weight loss and sarcopenia have detrimental effects on survival outcomes for GEA patients receiving perioperative FLOT [43,44]. These emphasise the importance of greater emphasis on dietary and rehabilitation support in the postoperative phase of treatment.

Our study has limitations due to the small sample size and retrospective nature of the analysis, with a relatively short duration of follow-up. In addition, only 14% of the patients were ≥70 years old. The small sample size is contributed to by the fact that only patients with more locally advanced disease were selected for perioperative treatment. Additionally, we do not have data on the impact on quality of life in our patients who undergo perioperative FLOT. The majority of our patients (78.6%) also had an ECOG status of 0. Nonetheless, we demonstrate that perioperative FLOT is tolerable in an Asian setting with low rates of neutropenia and neutropenic sepsis with prophylactic GCSF. In addition, despite selecting for patients with cT4 tumours or bulky lymph nodes, similar pathological complete response rates and survival outcomes were achieved compared to the FLOT4-AIO study.

## 5. Conclusions

This study presents a unique real-world Asian experience of perioperative FLOT with prophylactic GCSF use. The use of prophylactic GCSF reduced the rates of G3-4 neutropenia. Similar pathological complete response rates and survival outcomes were seen when compared to Western and real-world studies. We also found no differences in survival and toxicities between patients ≥ 65 years and <65 years, demonstrating the efficacy and tolerability of FLOT in elderly Asian patients. Patients of lower socioeconomic status, as represented by housing index, have poorer tolerability of FLOT, highlighting the importance of proactive measures to anticipate challenges and support vulnerable groups during treatment.

## Figures and Tables

**Figure 1 cancers-16-03697-f001:**
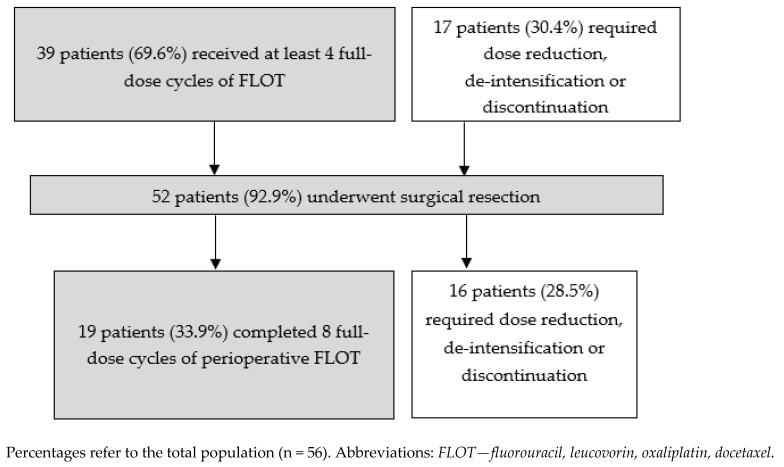
Treatment exposure (n = 56).

**Figure 2 cancers-16-03697-f002:**
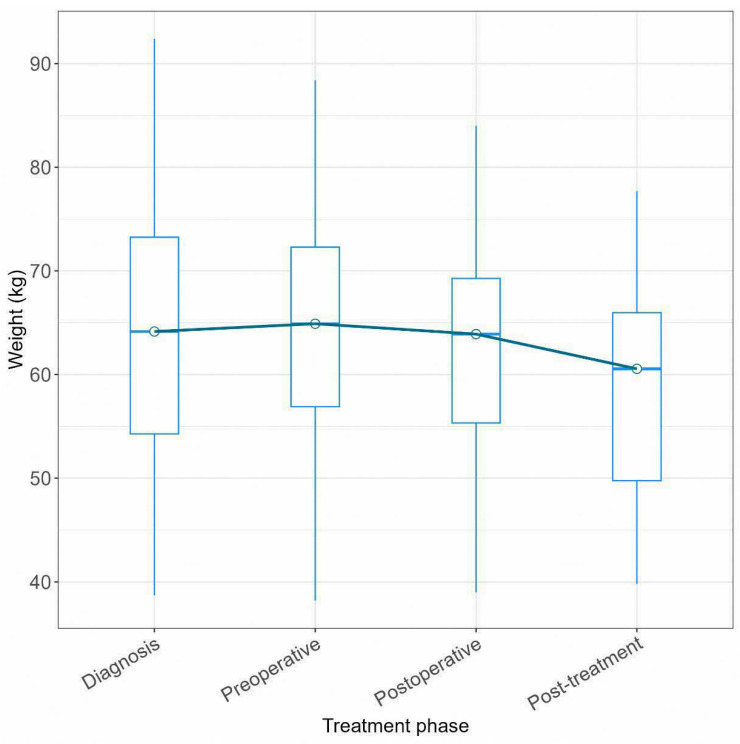
Median weight at different treatment phases.

**Figure 3 cancers-16-03697-f003:**
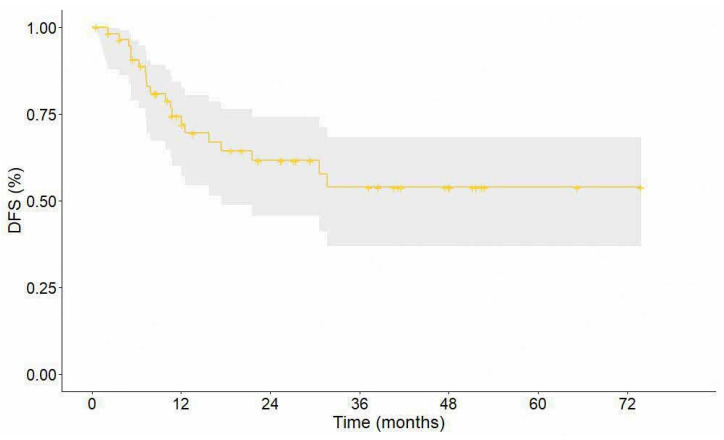
Disease-free survival. The grey area represents the 95% confidence interval.

**Figure 4 cancers-16-03697-f004:**
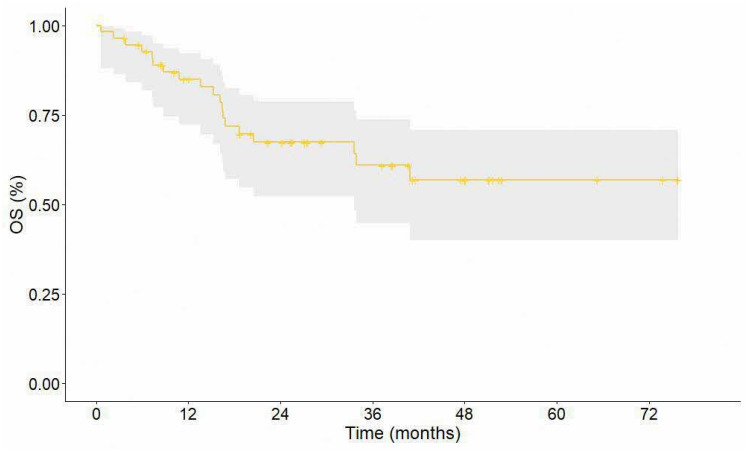
Overall survival. The grey area represents the 95% confidence interval.

**Table 1 cancers-16-03697-t001:** Baseline demographic and disease characteristics of a cohort (n = 56) of resectable GEA patients treated with perioperative FLOT in a real-world setting.

	Patients N (%)
Sex	
Male	47 (83.9)
Female	9 (16.1)
Age (years)	
Median (range)	64.0 (37.8–74.8)
<65 years	31 (55.4)
≥65 years	25 (44.6)
Race	
Chinese	48 (85.7)
Malay	3 (5.4)
Indian	5 (8.9)
Housing index	
Low (1–2)	5 (8.9)
Medium (3–4)	41 (73.2)
High (5–7)	10 (17.9)
ECOG PS	
0	44 (78.6)
1	12 (21.4)
BMI at diagnosis (kg/m^2^)	
<18.5	4 (7.1)
18.5–22.9	30 (53.6)
≥23.0	22 (39.3)
Location	
Stomach	23 (41.1)
GEJ	33 (58.9)
Siewert 1	9 (16.1)
Siewert 2	6 (10.7)
Siewert 3	4 (7.1)
Not specified/unknown	14 (25.0)
Signet ring cells	
Yes	12 (21.4)
Not specified	44 (78.6)
MMR status	
dMMR	2 (3.6)
pMMR	41 (73.2)
Not specified	13 (23.2)
HER2 status	
Positive	10 (17.9)
Negative	22 (39.2)
Unknown	24 (42.9)
cT	
T1	1 (1.8)
T2	4 (7.1)
T3	29 (51.8)
T4	22 (39.3)
cN	
N0	13 (23.2)
N1	23 (41.1)
N2	18 (32.1)
N3	2 (3.6)

Abbreviations: ECOG, Eastern Cooperative Oncology Group; PS, performance status; BMI, body mass index; MMR, mismatch repair.

**Table 2 cancers-16-03697-t002:** Chemotherapy-associated adverse events.

	Preoperative, N (%) (n = 56)		Postoperative, N (%) (n = 37)		Combined, N (%) (n = 56)	
	Grade 1–2	Grade 3–4	Grade 1–2	Grade 3–4	Grade 1–2	Grade 3–4
Haematologic						
Neutropenia	8 (14.3)	1 (1.8)	7 (18.9)	3 (8.1)	11 (19.6)	3 (5.4)
Febrile neutropenia	-	1 (1.8)	-	1 (2.7)	-	2 (3.6)
Anaemia	0	0	0	0	0	0
Gastrointestinal						
Diarrhoea	18 (30.5)	5 (8.9)	10 (27.0)	1 (2.7)	20 (35.7)	6 (10.7)
Nausea	15 (26.7)	1 (1.8)	5 (13.5)	0	18 (32.1)	1 (1.8)
Vomiting	12 (21.4)	3 (5.4)	4 (10.8)	0	14 (25.0)	3 (5.4)
Increased AST/ALT	1 (1.8)	0	1 (2.7)	0	1 (1.8)	0
Stomatitis/mucositis	2 (3.6)	0	1 (2.7)	0	3 (5.4)	0
Others						
Neuropathy	17 (30.4)	0	10 (27.0)	0	21 (37.5)	0
Infective event	0	0	1 (2.7)	1 (2.7)	1 (1.8)	1 (1.8)

**Table 3 cancers-16-03697-t003:** Univariate analysis for OS.

	All Patients (N = 56)			Surgical Population (N = 52)		
	Adverse Events/No	Univariable HR (95% CI)	*p*-Value	Adverse Events/No	Univariable HR (95% CI)	*p*-Value
Age						
<65	10/31	1		9/30	1	
≥65	9/25	1.08 (0.44–2.66)	0.869	7/22	0.98 (0.36–2.64)	0.968
Sex						
Male	18/47	1		15/43	1	
Female	1/9	0.28 (0.04–2.12)	0.219	1/9	0.33 (0.04–2.51)	0.284
Ethnicity						
Chinese	16/48	1		2/5	1	
Malay	1/3	0.81 (0.11–6.13)	0.837	1/3	0.94 (0.12–7.24)	0.952
Indian	2/5	1.55 (0.35–6.75)	0.561	13/44	1.94 (0.44–8.65)	0.383
BMI at diagnosis, kg/m^2^						
≥23	7/22	1		6/21	1	
18.5–22.9	10/30	1.10 (0.42–2.89)	0.851	8/27	1.07 (0.37–3.10)	0.894
<18.5	2/4	1.37 (0.28–6.60)	0.696	2/4	1.57 (0.32–7.80)	0.581
Housing index						
High (≥5)	5/10	1		4/9	1	
Medium (3–4)	11/41	0.60 (0.21–1.75)	0.353	10/39	0.72 (0.22–2.30)	0.576
Low (<3)	3/5	1.55 (0.37–6.55)	0.551	2/4	1.34 (0.24–7.36)	0.740
ECOG						
0	12/44	1		10/41	1	
1	7/12	2.22 (0.87–5.65)	0.093	6/11	2.24 (0.81–6.18)	0.118
Tumour site						
GEJ	13/33	1		12/31	1	
Stomach	6/23	0.50 (0.19–1.33)	0.164	4/21	0.33 (0.11–1.04)	0.058
MMR						
pMMR	13/41	1		11/38	1	
dMMR	1/2	2.58 (0.33–20.43)	0.370	1/2	3.42 (0.42–28.15)	0.253
Lauren’s type						
Intestinal	5/13	1		4/12	1	
Diffuse	0/2	NE		0/2	NE	
Mixed	1/6	0.26 (0.03–2.29)	0.227	1/6	0.30 (0.03–2.72)	0.286
Not evaluable	13/35	0.78 (0.27–2.20)	0.635	11/32	0.80 (0.25–2.53)	0.704
Signet ring cells						
Not specified	17/44	1		15/41	1	
Yes	2/12	0.43 (0.10–1.88)	0.264	1/11	0.24 (0.03–1.81)	0.166
cN						
N0	4/13	1		3/12	1	
N1–N3	15/43	0.99 (0.33–2.98)	0.981	13/40	1.14 (0.32–4.00)	0.840
Clinical stage						
II	3/15	1		3/15	1	
III	16/41	2.13 (0.62–7.32)	0.229	13/37	1.80 (0.51–6.33)	0.358
Grading						
3	9/27	1		8/26	1	
2	7/11	1.72 (0.64–4.62)	0.283	7/11	1.91 (0.69–5.26)	0.213
1	0/5	NE		0/5	NE	
4 cycles full-dose FLOT						
No	10/30	1		8/28	1	
Yes	9/26	1.05 (0.43–2.58)	0.920	8/24	1.22 (0.46–3.26)	0.687
8 cycles full-dose FLOT						
No	12/35	1		9/31	1	
Yes	7/21	0.86 (0.34–2.19)	0.753	7/21	1.09 (0.40–2.92)	0.868
Significant weight loss (>10%)						
No	15/39	1		12/35	1	
Yes	4/17	0.50 (0.17–1.51)	0.217	4/17	0.59 (0.19–1.82)	0.356
Resection margin						
R1				6/10	1	
R0				9/39	0.17 (0.06–0.50)	0.001
R2				1/1	2.47 (0.27–22.65)	0.425
ypT						
T1–T4				15/44	1	
T0				1/8	0.35 (0.05–2.69)	0.316
ypN						
N1–N3				14/30	1	
N0				2/22	0.14 (0.03–0.64)	0.011
Pathological stage						
0–II				6/30	1	
III–IV				10/22	3.63 (1.30–10.12)	0.014
pCR						
No				15/44	1	
Yes				1/8	0.35 (0.05–2.69)	0.316
Major postoperative complications						
3 and above				7/13	1	
Nil				8/38	0.30 (0.11–0.84)	0.022
ASA score						
2				7/27	1	
3				9/25	2.34 (0.86–6.37)	0.097
AJCC/CAP tumour regression grading						
3				4/10	1	
2				10/25	0.51 (0.16–1.67)	0.266
1				0/2	NE	
0				1/8	0.17 (0.02–1.49)	0.109
0				1/8	0.14 (0.02–1.31)	0.086

## Data Availability

The original contributions presented in this study are included in the article/Appendix A. Further inquiries can be directed to the corresponding author.

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
