# Peer review of "Real-World Outcomes for Localised Gastro-Oesophageal Adenocarcinoma Cancer Treated with Perioperative FLOT and Prophylactic GCSF Support in a Single Asian Centre"

_cancers, 2024, doi:10.3390/cancers16213697_

Round 1
Reviewer 1 Report
Comments and Suggestions for Authors
In this manuscript, the authors present the "Real-world outcomes for localized gastro-oesophageal adenocarcinoma treated with perioperative FLOT and prophylactic G-CSF support in a single Asian center." While the results appear intriguing, the topic lacks novelty and requires further refinement. My detailed feedback is outlined below:
-
The research focuses on gastro-oesophageal adenocarcinoma (GEA) patients, encompassing 23 stomach cancer cases and 33 gastroesophageal junction (GEJ) cancer cases. However, it is well-established that the treatment strategies for stomach cancer and GEJ cancer differ significantly based on current guidelines. To enhance the clarity and focus of the study, it would be advisable to concentrate on one type of GEA cancer.
-
One limitation of this paper is the small sample size, which includes only 56 patients. Additionally, the retrospective nature of the research further restricts its generalizability.
-
Regarding prophylactic G-CSF support, it would be beneficial to specify the type of G-CSF used, whether long-acting or short-acting drugs. Furthermore, a comparative analysis of adverse events (AEs) between patients receiving G-CSF and those without G-CSF in the same Asian center would provide valuable insights.
-
Overall, the English language used in the manuscript is acceptable, but minor editing could enhance its clarity. For instance, Figure 1 does not display the complete form, and in Table 3, "Event/No" could be more clearly described as "Adverse Events/No."
The English language used in the manuscript is acceptable, but minor editing could enhance its clarity. For instance, Figure 1 does not display the complete form, and in Table 3, "Event/No" could be more clearly described as "Adverse Events/No."
Author Response
- Comment 1: The research focuses on gastro-oesophageal adenocarcinoma (GEA) patients, encompassing 23 stomach cancer cases and 33 gastroesophageal junction (GEJ) cancer cases. However, it is well-established that the treatment strategies for stomach cancer and GEJ cancer differ significantly based on current guidelines. To enhance the clarity and focus of the study, it would be advisable to concentrate on one type of GEA cancer.
- Response 1: We thank the reviewer for the valuable comments. We agree that there can be different preferred strategies for gastric cancer and GEJ cancers, especially in Asia where upfront surgery is generally preferred for gastric cancers and Type 3 GEJ cancers and in Western guidelines where pre-op chemoradiation is an alternative in Type 1 and 2 GEJ cancers. As the reviewer has pointed out that our study has a limited sample size, hence we feel that focusing only either gastric or GEJ would dilute the results. Furthermore, in our univariate analysis, we did not show a significant difference in outcomes for gastric versus GEJ cancers for both OS (Table S6) and PFS (Supplementary Table ) Our study reflects a real world approach where treatment strategies are personalized based on MDT discussions to the optimal strategy for these patients. Hence most of our recruited gastric cancer patients in this study were evaluated at tumour board to have locally advanced tumours which are either cT4 or bulky nodal disease and recommend to have peri-operative FLOT rather than upfront gastrectomy. Such stratification is also recognized in the Korean and Japanese guidelines which indicate that pre-operative chemotherapy can be considered. We have referenced this is in our introduction (Japanese guidelines 11, Korean guidelines 12. With respect to pre-operative chemoradiation as an alternative, the recently present ESOPEC study 10 has demonstrated the superiority of FLOT over CROSS chemoradiation in Type 1 and 2 GEJ hence highlighting the relevance and increasing importance of FLOT for GEJ cancers as well as in gastric cancers. This is also reflected in the NCCN and ESMO guidelines. Therefore, we attempt to show here the tolerability and efficacy of the FLOT regimen for GEA tumours in an Asian population, which can then be contrasted to studies of more commonly use peri-operative regimens in Asia such as the DOS regimen. Furthermore, we refer to the pivotal perioperative chemotherapy studies of AIO-FLOT 4 and MAGIC where both gastric and GEJ tumours were included.
- Comment 2: One limitation of this paper is the small sample size, which includes only 56 patients. Additionally, the retrospective nature of the research further restricts its generalizability.
- Response 2: We acknowledge our limited sample size as in point 1. We have taken steps to reduce bias by including all patients who were treated with FLOT during the study period and majority of patients had consented for prospective data collection as part of an ethics approved protocol.
- Comment 3: Regarding prophylactic G-CSF support, it would be beneficial to specify the type of G-CSF used, whether long-acting or short-acting drugs. Furthermore, a comparative analysis of adverse events (AEs) between patients receiving G-CSF and those without G-CSF in the same Asian center would provide valuable insights.
- Response 3: We have provided the type of GSCF used in Section 2.2 Line 105 as well as Section 3.2 line 173-174.
- Comment 4: Overall, the English language used in the manuscript is acceptable, but minor editing could enhance its clarity. For instance, Figure 1 does not display the complete form, and in Table 3, "Event/No" could be more clearly described as "Adverse Events/No."
- Response 4: We have amended this as suggested.
Reviewer 2 Report
Comments and Suggestions for Authors
Significant study conducted by Asian colleagues, who took into consideration the FLOT treatment, typically used in European countries, for the treatment of gastric cancer and gastroesophageal junction. We absolutely agree with their diagnostic framework and with the discussion in the multidisciplinary commission to best frame the pathology and therefore follow the best therapeutic treatment. We also recommend that they study already in the biopsies: HER 2 and microbiology that can better guide the choice of neoadjuvant therapy. Very precise tables from which all the data relating to the pathology and the FLOT cycles are deduced. It would be useful in the histological examination to know what percentage of signet ring cells were in the neoplasia to have a predictive data. Nothing to say about the statistics. For what is written on the discussion we can only point out that there is also in the literature the DOC doi.org/10.1016/j.suronc.2019.10.002, to be cited in the literature, same drugs as FLOT administered by another route, with similar effects, for which fewer adverse effects have been noted and reported. Excellent iconography, good English, good bibliography on which the entire work is based
Author Response
Comment: Significant study conducted by Asian colleagues, who took into consideration the FLOT treatment, typically used in European countries, for the treatment of gastric cancer and gastroesophageal junction. We absolutely agree with their diagnostic framework and with the discussion in the multidisciplinary commission to best frame the pathology and therefore follow the best therapeutic treatment. We also recommend that they study already in the biopsies: HER 2 and microbiology that can better guide the choice of neoadjuvant therapy. Very precise tables from which all the data relating to the pathology and the FLOT cycles are deduced. It would be useful in the histological examination to know what percentage of signet ring cells were in the neoplasia to have a predictive data. Nothing to say about the statistics. For what is written on the discussion we can only point out that there is also in the literature the DOC doi.org/10.1016/j.suronc.2019.10.002, to be cited in the literature, same drugs as FLOT administered by another route, with similar effects, for which fewer adverse effects have been noted and reported. Excellent iconography, good English, good bibliography on which the entire work is based
Response:
- We thank the reviewer for the valuable and positive comments. Previous IHC MMR status was provided, we have now included the HER2 status and frequency of signet ring cells in Table 2. Only 12 patients had signet rings in the histology which does not allow us to make any meaningful comparisons. We do not test EBV status in the tumour routinely and do not have this information available.
- We have referenced the DOC data as suggested in the Discussion Reference 42.